# Topologically-imposed vacancies and mobile solid ³He on carbon nanotube

I. Todoshchenko [1] ✉, M. Kamada[1,2], J.-P. Kaikkonen[1], Y. Liao[3], A. Savin [1,2], M. Will [1,2], E. Sergeicheva[1], T. S. Abhilash[1], E. Kauppinen [3] & P. J. Hakonen [1,2] ✉

Low dimensional fermionic quantum systems are exceptionally interesting because they reveal distinctive physical phenomena, including among others, topologically protected excitations, edge states, frustration, and fractionalization. Our aim was to confine ³He on a suspended carbon nanotube to form 2-dimensional Fermi-system. Here we report our measurements of the mechanical resonance of the nanotube with adsorbed sub-monolayer down to 10 mK. At intermediate coverages we have observed the famous 1/3 commensurate solid. However, at larger monolayer densities we have observed a quantum phase transition from 1/3 solid to an unknown, soft, and mobile solid phase. We interpret this mobile solid phase as a bosonic commensurate crystal consisting of helium dimers with topologically-induced zero-point vacancies which are delocalized at low temperatures. We thus demonstrate that ³He on a nanotube merges both fermionic and bosonic phenomena, with a quantum phase transition between fermionic solid 1/3 phase and the observed bosonic dimer solid.

Physics of quantum liquids and solids is eminently rich and often counterintuitive. Bosonic liquid ⁴He possesses superfluidity, e.g., ability to flow without resistance, due to Bose-Einstein condensation of particles below 2.1 K[1,2]. Its fermionic counterpart, liquid ³He displays superfluidity below 2.5 mK due to paring of atoms into bosonic dimers (so-called Cooper pairs)[3,4]. Moreover, a notion has been put forward that bosonic quantum solid could behave like a fluid, due to mobile delocalized defects such as vacancies. Yet, those vacancies, if present at low enough temperature, must Bose-condense, and the crystal must then behave as a superfluid, keeping at the same time its crystalline translational symmetry[5–7]. Search for this spectacular supersolid state has been ongoing since 1980s, and Kim and Chan announced hints of superflow in solid ⁴He in 2004[8,9]. However, later it was established that the equilibrium concentration of vacancies is exponentially small at low temperatures[10,11], and the observed superflow was most likely through superfluid inside cores of dislocations or on grain boundaries[12]. Thus, it is extremely hard to

observe a superflow even in the most quantum crystal, solid helium, in three dimensions.

Lowering the dimensionality, however, opens up the possibility for topologically stabilized vacancies. In particular, carbon nanotube (CNT), which can be viewed as a rolled-and-glued narrow strip of graphene (a sheet of c-plane of graphite), has generally a symmetry that does not include symmetry of commensurate superlattice of adsorbed helium. This mismatch ensures a finite amount of defects/vacancies in adsorbed solid helium even at zero temperature, i.e., so-called zero-point vacancies, which give their hallmark on the ground state, and which have not yet been ever found in the bulk 3D crystals. In the planar geometry of graphite/grafoil where carbon atoms are assembled into a hexagonal lattice, the famous 1/3 (every third carbon hexagon is occupied by helium atom) commensurate solid phase is very well established[13–17]. Another commensurate phase[18] is stabilized at densities corresponding to the dimer 2/5 solid suggested by Greywall[19]. On a curved nanotube the dimer commensurate phase

¹Low Temperature Laboratory, Department of Applied Physics, Aalto University School of Science, P.O. Box 15100, FI-00076 Espoo, Finland. ²QTF Centre of Excellence, Department of Applied Physics, Aalto University, P.O. Box 15100, 00076 Espoo, Finland. ³Nano Materials Group, Department of Applied Physics, Aalto University School of Science, P.O. Box 13500, FI-00076 Espoo, Finland. ✉e-mail: igor.todoshchenko@aalto.fi; pertti.hakonen@aalto.fi

should be even more stable due to a larger distance between adsorption cites.

The dimers are stabilized by the carbon lattice, despite a $He_2$ molecule by itself has very low, about 1 mK, binding energy[20] owing to the weakness of the van der Waals attraction and strong zero-point motion. The zero-point energy can be estimated as the localization energy of a particle with reduced mass $m_3/2$ in a box of size $d = 2.5$ Å (distance between neighboring sites on the carbon lattice), $\pi^2\hbar^2/(m_3d^2) = 25$ K which is much larger than van der Waals minimum 10 K of He-He interaction potential[21]. On its part, carbon lattice produces for each helium atom a modulation of adsorption potential with the amplitude of ~30 K[22]. In total, the potential of the carbon lattice enhanced by the He-He interaction is strong enough to stabilize two helium atoms, a dimer, in the centers of neighboring hexagons. Missing dimers, or vacancies, in a bosonic dimer crystal are also bosonic, and delocalize due to strong zero-point motion to procure a mobile, fluid-like solid phase which we present in this work. The great importance of topological frustrations in low dimensionality has recently been demonstrated also in nanomagnetism[23].

Helium on graphite/grafoil has been investigated extensively by heat capacity, NMR, and by neutron scattering measurements[16]. A smallness of helium sample, ~$10^4$ atoms, makes similar measurements extremely difficult in the case of a nanotube. Instead, we were measuring mechanical characteristics of a nanotube with adsorbed helium, which also gave assess to thermodynamical properties. Extraordinary electrical and mechanical properties of suspended CNTs facilitate their use as ultra-sensitive detectors for force sensing[24] and for surface physics. Pioneering work by Cobden et al. and Lee et al. have shown suspended nanotube to be extremely sensitive tool to probe the properties of adsorbed noble gases[25,26]. An oscillating carbon nanotube was employed by Noury et al. to investigate layering transitions and superfluidity of single/multiple layers of adsorbed $^4$He on its surface[27].

In this work we have measured the mechanical and thermodynamic properties of $^3$He adsorbed on a suspended carbon nanotube. We demonstrate that at high densities monolayer solid $^3$He softens dramatically and behaves like a fluid. We attribute the observed phenomenon to delocalized topologically protected vacancies.

## Results

### Observation of transitions

A low-coverage $^3$He layer on CNT displayed a clustered liquid + dilute gas to uniform fluid transition at temperatures 0.1,...,0.5 K. This transition manifested itself as an abrupt drop of the CNT resonant frequency $F_0$ with increasing temperature $T$, due to redistribution of helium mass from the ends to uniform coverage over the tube [see Fig. 1a]. At larger coverages the observed jumps in $F_0(T)$ traces [Fig. 1b, c] are identified as changes in a stiffness of the oscillator due to melting. In the $F_0(T)$ trace at large coverage, Fig. 1f, there are two abrupt frequency changes, positive and negative, corresponding to modifications of stiffness across the structural phase transition. The 1/3 and the 2/5 commensurate solid phases are illustrated in the inserts of Fig. 2.

Additional stiffness due to commensurate solid helium appears as a result of interaction between carbon and helium atoms[28]. The vertical projection of the force acting on individual carbon atom from helium atom placed above the center of hexagon is $f_\perp = (1/6) dU/dz$ where $U(z)$ is the adsorbtion potential. The longitudinal force is then $f_\parallel = (a/6z_0)dU/dz|_{z_0} = 1$ pN where $z_0 = 2.88$ Å is the altitude of helium atom[16,22], and $dU/dz|_{z_0} \simeq 80$ K/Å[22]. Additional tension is the sum of the individual forces over the circumference, $\delta\mathcal{F} = 2\pi r/(3\sqrt{3}a)2f_\parallel = 14$ pN for the 1/3 solid. This constitutes about 2% of tension of the bare nanotube, $\mathcal{F}_0 = 4F_0^2L^2\mu \simeq 800$ pN, where $L = 700$ nm and $\mu = 3.8\cdot10^{-15}$ kg/m are length and mass density, respectively. The increase of the resonance frequency due to 1/3 solid is thus estimated as ~1% which is in fairly good agreement with the experiment. Gas or liquid helium, for all processes slower than the tunneling frequency, can be viewed as a uniform layer which does not produce additional tension because of the symmetric distribution around any individual carbon atom.

### Phase diagram

The phase diagram, constructed from $F_0(T)$ sweeps, is displayed in Fig. 2. Despite the strong (tens of Kelvin) adsorption potential modulation along the carbon lattice[29], due to the high probability of tunneling, an individual atom is moving almost freely, see the Methods section. Indeed, at low coverages gas and liquid helium are formed on

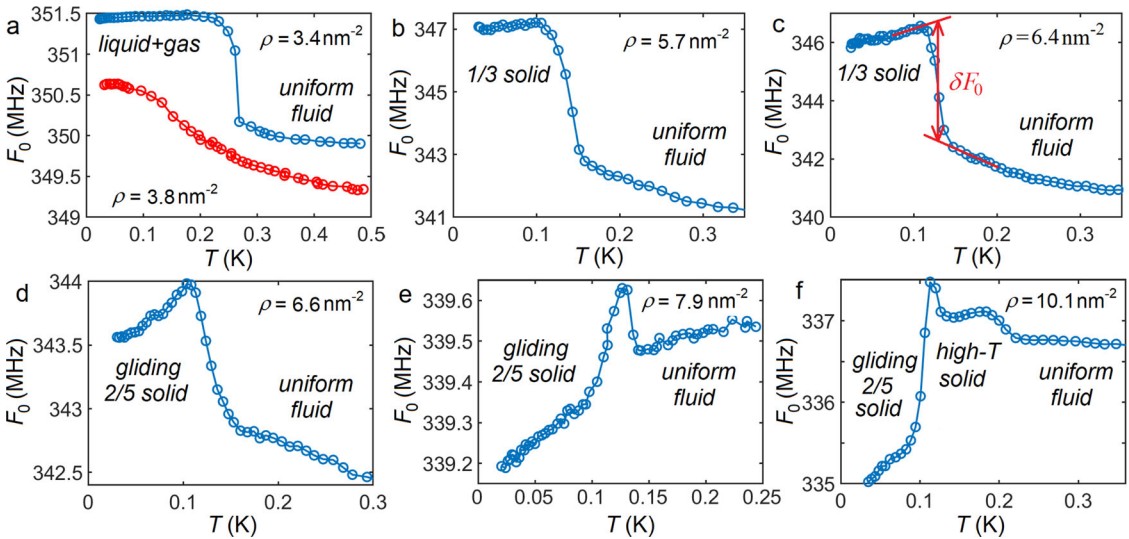

**Fig. 1 | Phase transitions observed as a jump of the resonant frequency $F_0$ at different helium coverages. a** Transitions from the liquid+gas coexistence to a uniform fluid. The jump of $F_0$ is due to redistribution of helium from liquid clusters to uniform. **b**, **c** Melting of the rigid 1/3 solid manifested by the reduction of the stiffness of the oscillator. The panel **c** also illustrates the metering of the jump $\delta F_0$ at the transition. **d**, **e** Melting of the discovered gliding solid phase with no measurable jump of the resonance frequency $F_0$ across the transition. **f** Transition from gliding to the rigid high-$T$ solid manifested by an abrupt increase of the stiffness of the oscillating tube. The second transition from high-$T$ solid to liquid is seen as a drop of the stiffness, as in **b**, **c**. The nearly linear $F_0(T)$ dependence below 0.1 K can be extrapolated to the liquid, with no noticeable jump, as in panel **e**.

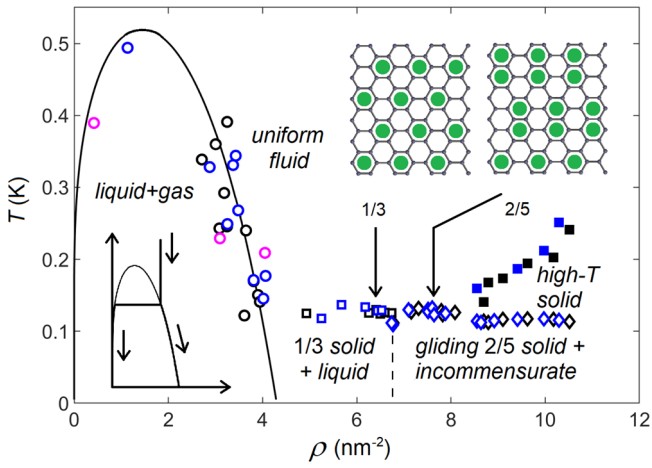

**Fig. 2 | Overall phase diagram of ³He sub-monolayer on a carbon nanotube.** The individual points on the phase diagram were obtained from $F_0(T)$ traces illustrated in Fig. 1. Circles: transitions from the phase-separated clustered liquid + gas system to uniform fluid; open squares: transitions from the mixture 1/3 solid plus liquid to uniform fluid; diamonds: transition from the gliding 2/5 solid to fluid below 8.2 nm⁻² and to high-*T* solid at larger coverages; filled squares: transition from the high-*T* solid to liquid. Different colors represent different mechanical resonances. The solid curve depicts the fit of the van der Waals liquid-gas phase separation curve (see the Methods section for details). The transition occurs at temperature $T_c(\rho)$ where the line $\rho = \mathrm{const}(T)$ crosses the phase separation curve, see the insert on the bottom left. Two other inserts and vertical arrows show the coverages and structures of the 1/3 and 2/5 commensurate solids on grafoil.

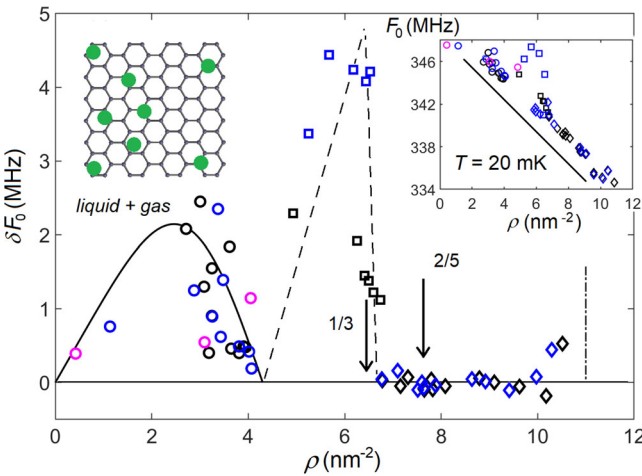

**Fig. 3 | Excess $\delta F_0$ of the resonant frequency in the low-temperature phases compared to the liquid, see Fig. 1(c).** Circles: transition from clustered liquid + gas coexistence to a uniform fluid. The $\delta F_0(\rho)$ dependence is described very well by Eq. (3) in the Methods section (solid curve), see text. Squares: melting of the 1/3 solid. The dashed lines indicate the anticipated fraction of the 1/3 phase. Diamonds: difference between gliding solid and fluid (at $\rho > 8.2$ nm⁻² the resonant frequency $F_0$ was extrapolated linearly from gliding solid through high-*T* solid phase). Absence of the drop of the resonant frequency on melting indicates that there is no additional stiffness provided by gliding solid. The insert on the left illustrates liquid-gas coexistence. The insert on the right shows quantum phase transitions from fluid to the 1/3 solid and from the 1/3 solid to the soft gliding solid. Dash-dotted line: beginning of the second layer promotion on grafoil[14,33].

graphite[30]. In our experiments, the transition temperature $T_c$ has a maximum at around $\rho \sim 1.5$ nm⁻², which is a characteristic feature of the liquid-gas separation. The van der Waals (vdW) approach adapted for the 2D case (see the Methods section) describes the observed separation temperature and the value of the jump very well, as can be seen from Figs. 2 and 3. A fit of the calculated 2D vdW phase separation curve to the measured transition temperatures is illustrated by the parabola-looking curve in Fig. 2 at densities below 4.2 nm⁻². The jump at the separation temperature occurs due to re-distribution of helium from the clusters at the ends of the tube to a uniform layer, which enhances the effective mass of the oscillator. After the critical coverage $\rho_c = 1.4$ nm⁻² and critical temperature $T_c = 0.52$ K are determined from the fit shown in Fig. 2, the magnitude (3) of the jump $\delta F_0$ can be calculated to be in a very good agreement with experimental data, as shown in Fig. 3.

When the liquid state fully overtakes the gas phase at 4.2 nm⁻², the jump $\delta F_0$ approaches zero, but with further increase of coverage there emerges again a large frequency drop (Figs. 1b, c and 3). This frequency decrease indicates a transition from solid to fluid, as solid helium phase enhances the tension in the tube (see above). The coverage 6.37 nm⁻² corresponds to the 1/3 commensurate phase (see the insert in Fig. 2) which maximizes the additional tension and thereby the frequency jump $\delta F_0$, as seen in Fig. 3. The melting temperature of the 1/3 phase on CNT, $T_c \approx 0.13$ K in Fig. 2, is suppressed in comparison with the $T_c \approx 3$ K on graphite[13,18]. This is a clear signature of larger distance between ³He atoms and carbon atoms on a CNT due to its curvature[31], which weakens the He-C interaction. This observation agrees well with the reduction of adsorption energy from ~140 K on graphite[22] to ~100 K on CNT[31].

The frequency change $\delta F_0$ across the melting temperature disappears at coverages $\rho \approx 6.7$ nm⁻² (Fig. 1), which means that the additional tension due to the solid helium phase vanishes. We attribute such a soft solid state to a mobile solid phase containing a gas of delocalized vacancies. This solid with mobile vacancies does not produce additional tension on the tube, likewise a liquid, because in these

phases helium atoms are delocalized, and helium density distribution is symmetric near each carbon atom, resulting in zero net force. This interpretation is corroborated by the observation of two transitions at higher coverage ≳8.4 nm⁻² shown in Fig. 1f: the solid-induced tension appears abruptly at $T \approx 0.1$ K (mobile gliding solid to high-*T* solid transition), and then disappears again with melting of high-*T* solid at $T = 0.13,...,0.25$ K depending on the coverage. The frequency jump $\delta F_0$ between 1/3 phase and the gliding solid is observed down to the lowest temperatures, as shown in the insert in Fig. 3, which indicates a *quantum phase transition* in the adsorbed film as a function of coverage.

We interpret the transition from rigid solid to gliding solid at ~6.7 nm⁻² as a transformation of rigid 1/3 fermionic solid into a bosonic dimer solid with delocalized vacancies. According to Andreev[32], in a magnetically disordered fermionic solid ³He a vacancy, due to its magnetic moment, forms a ferromagnetic polaron because of the enhanced (ferromagnetic) pair exchange between helium atoms in its vicinity. The motion of such vacancy must be accompanied with a spin current, while in the disordered paramagnetic solid, in the absence of magnons, there is no effective mechanism of spin transport. A vacancy in fermionic solid is therefore immobile[32]. Indeed, the monomer fermionic 1/3 phase shows no signs of mobile vacancies. In contrast, vacancies in the bosonic dimer 2/5 solid phase carry no spin and should become delocalized at low enough temperatures. Although helium on graphite/grafoil forms a triangular incommensurate solid at elevated densities[14,16,33], it will be reasoned in the Methods section that on a nanotube the dimer 2/5 phase is stabilized instead. Due to the anti-symmetrical wavefunction of dimer formed by two identical fermions, and due to the symmetric rotational wavefunction, $J = 0$, spins of atoms in the dimer are anti-parallel, and the dimer is a boson with $S = 0$ (see the Methods section).

## Elementary excitations

The temperature dependence of the resonant frequency $F_0(T)$ in low temperature phases was also investigated as a function of ³He

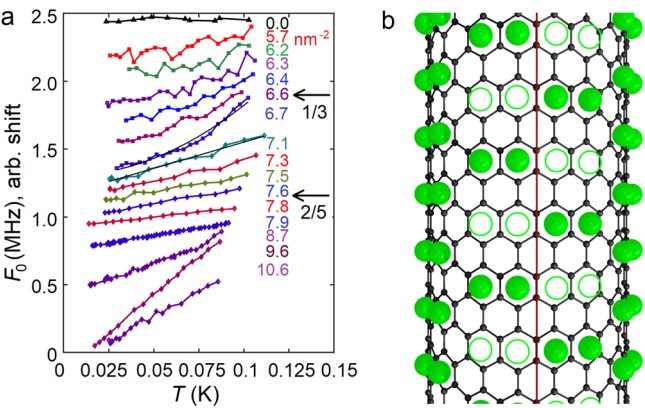

**Fig. 4 | Vacancies on a nanotube. a** Temperature dependence of resonant frequency $F_0$ at different coverages. Curves are shifted arbitrarily in the vertical direction. We find $F_0(T)-F_0(0) \propto T^\beta$ with $\beta \simeq 2$ at coverages $\leqslant 6.7\,nm^{-2}$, which corresponds to 1D phonons in a solid. The quadratic behavior changes abruptly to a linear at $\rho \approx 7.1\,nm^{-2}$, which persists till complete monolayer coverage. Linear temperature dependence can be obtained with constant number of weakly interacting excitations, such as vacancies induced by topological mismatch. The quadratic fit for the highest coverage $6.7\,nm^{-2}$ of the 1/3 solid, and the linear fit for the lowest coverage $7.1\,nm^{-2}$ of gliding solid are shown by black curves. **b** Illustration of mismatch-induced defects in high-coverage $^3$He superlattice adsorbed on a CNT. The defects are produced when a graphene sheet with helium superlattice on it is rolled around the vertical axis to form a CNT. The dimerized 2/5 superlattice is depicted as an example. The red line shows the gluing of edges of the graphene stencil. Empty circles illustrate vacancies created by the mismatch.

coverage [see Fig. 4a]. The helium pressure, which is the net force acting on the ends of the tube from helium particles, contribute to the tension $\mathcal{F}$ of the bare tube. Therefore, change of resonance frequency of a strained tube $F_0 = (1/2L)\sqrt{\mathcal{F}/(ML)}$ reflects the change in helium pressure $P$. The power-law fitting of the data, $F_0(T)-F_0(0) = aT^\beta$ gave $\beta = 2.1 \pm 0.1$ in the 1/3 solid below $7.1\,nm^{-2}$, corresponding to 1D phonons, see the Methods section. In helium adsorbed on graphite, apart from nuclear magnetism in $^3$He, 2D phonons were shown to be dominating excitations at low temperatures[13,33]. On nanotube the orbital phonons have energies larger than $\hbar c/r \sim 1\,K$ ($c \simeq 200$ m/s is the speed of sound[13]) and thus frozen at our temperatures leaving only (1D) longitudinal phonons, which we indeed observe. The difference in coefficient $a_{1/3}$ is likely a manifestation of the size effect: the longer the 1/3 phase, the more phonon modes at smaller and smaller angles are excited at the same temperature. Indeed, maximum of $a_{1/3}$ is almost exactly at the 1/3 filling.

In the gliding solid phase, we find the exponent $\beta = 1.00 \pm 0.05$. The observed linear $P(T)$ dependence implies ideal gas type of behavior with a fixed number of particles which can be related to delocalized zero-point vacancies[5]. Truly, the nanotube as a seamlessly rolled graphene sheet works as a template for a superlattice of $^3$He dimers. However, helium superlattice has a larger period and cannot generally match the CNT lattice, therefore helium lattice contains finite number of vacancies, as shown in Fig. 4(b) for the 2/5 phase. From the topological point of view, the vacancies are stablized by the discrepancy between chirality [$NM$] of the nanotube, and the translational vector [$nm$] of the particular helium lattice. For the simplest case of the 1/3 commensurate solid with translations $t_1 = [1\,1]$ and $t_2 = [2\,-1]$, the discrepancy $D = \min([NM] - i \cdot t_1 - j \cdot t_2)$ can be considered as the topological invariant. Indeed, the chirality of the tube cannot be changed by the deformation only, without cutting/glueing. Hence, the invariant $D$ which determines and conserves the number of zero-point defects cannot be changed due to the mutual motion of carbon and helium lattices. The same is valid for any commensurate helium solid; for instance, 2/5 has $t_1 = [2\,1]$ and $t_2 = [3\,-1]$, and has the discrepancy $D = [0\,2]$ with the semi-metallic, nearly armchair tube [22 1]. Such a

discrepancy provides and stabilizes two single vacancies per one carbon ring, or one dimer vacancy. The system of vacancies is thus topologically protected by the mismatch, and no new defects appear with increasing temperature as the corresponding activation energy is of the order 10 K[10,11]. The variation of the slope $a_{2/5}$ involves also a size effect as the smallest slope $a_{2/5}$ corresponds nearly to the 2/5 coverage where the length occupied by this phase is the largest. The reason for the size effect could be multiple scattering of vacancies on both boundaries with the scattering frequency inversely proportional to the square of the length $L_{2/5}$.

## Discussion

Delocalized topological zero-point vacancies in the gliding solid phase become localized at some finite temperature[5]. Transition from gliding solid to rigid high-$T$ solid would then take place at temperature corresponding to the energy of localization of a vacancy within a length shorter than the inter-vacancy distance $D$, $k_B T_{loc} = \pi^2 \hbar^2/(2m^* D^2)$. We should stress that in this coarse estimation of localization temperature we have ignored the entropy contribution to the free energy of phases, which may affect significantly the value of $T_{loc}$. However, the detailed thermodynamical consideration of localization-delocalization transition is beyond the scope of the present paper, although this question is of great theoretical interest.

Elastic interaction between two individual vacancies leads to the repulsion, because the (quadratic) energy of deformation due to two distant vacancies is twice smaller than that due to two closely situated vacancies. Due to the mutual repulsion, vacancies distribute uniformly over the tube rather than form vacuum clusters/islands. The average distance $D$ between vacancies is thus set by the template of the particular type of the mismatch. The change of the total coverage only changes the relative fraction of the commensurate solid phase (see the Methods section) but does not change $D$ and, consequently, the transition temperature.

Assuming triangular lattice of vacancies, one can calculate the distance between neighboring vacancies in the 2/5 solid as $D = \sqrt{2\sqrt{3}\pi h a}$ where $h = 11$ Å is the distance from the axis to helium atoms. For the particular mismatch illustrated in Fig. 4b, $D = 1.8$ nm which gives $T_{loc} \approx 0.12$ K with the effective mass $m^* = 2m_3$ ($m_3$ is $^3$He atomic mass). Taking into account that, due to interaction, the effective mass should be larger than the bare mass, the estimated localization temperature fits the observed transition temperature very well.

The model of mobile delocalized vacancies was also supported by measurement of dissipation in different phases. The width of the resonance of the bare nanotube is 0.08-0.1 MHz, and it increases, roughly linearly, with the helium coverage. There is, however, a distinct difference between dissipation in different phases: fluid always provides a larger dissipation than the rigid solid due to the inertia: the fluid lags behind the motion of the tube, which results in (dissipative) relative motion of CNT and helium, while in the rigid 1/3 and in the high-$T$ solids this effect is much weaker. However, the dissipation in the discovered gliding solid phase is stronger than even in the fluid, albeit the temperature is lower. This can only be explained by yet larger mobility $\mu \equiv \dot{x}/f$ ($f = m\ddot{x}$ is the inertial force), providing larger relative velocity $v$ and consequently larger dissipation power $\mathcal{P} = \dot{x}f = \mu f^2$.

A distinctive peak in the resonance frequency at the transition from gliding solid to higher temperature phases [Fig. 1(d-f)] can be attributed to critical fluctuations. We note first, that in a monolayer of adsorbed atoms all transitions in an adsorbed monolayer are of the second or higher order. Indeed, at a first order transition all thermodynamical quantities, density in particular, experience a jump. In contrast, adsorbed atoms occupy the whole available surface (except for small coverages), and therefore there is no change in the density over the transition. Bretz et al. were first to confirm that in a melting of helium on grafoil is indeed a second order transition characterized by the fluctuation peak in the specific heat[13]. The (exact) theoretical

calculations for the 2D systems with nearest-neighbor repulsion also show second-order solution[34–37]. Critical fluctuations near the second order transition lead to the divergence of the heat capacity[38], and hence of the thermal expansion coefficient. As the density remains constant, the peak in thermal expansion coefficient leads to a peak in pressure $P$, and consequently in tension of the tube and in the resonant frequency $F_0$. The absence of hysteresis, except for low coverages, also points to a second order transition.

A crystal with delocalized vacancies possesses, at the same time, a translational symmetry like usual solid, and fluid-like mass flow due to delocalized vacancies. In particular, such a fluid-like solid cannot support gradient of pressure: a flow of vacancies will change the average density of particles to compensate it[5]. The system of bosonic vacancies in a quantum crystal can be therefore considered as a weakly interacting Bose-gas which should condense to the single ground state at low enough temperature[5]. Such a hypothetic state is called supersolid which, preserving a crystalline symmetry, possesses a superflow of mass and other properties of a superfluid. Let us estimate the Bose-condensation temperature for our system of topologically stabilized zero-point vacancies in solid $^3He$ on nanotube.

The transversal motion of an individual vacancy around the tube has an energy $\varepsilon_\phi = \hbar^2 l^2/(2m^* r^2)$ where $l$ is the orbital quantum number. The first excited state with $l = 1$ and $m^* = 2m_3$ has an energy of 60 mK, and therefore at lower temperatures the transversal motion is frozen, and the system of vacancies becomes truly one-dimensional. One can estimate the Bose-Einstein condensation temperature of 1D system of vacancies by adapting the standard procedure (see for instance[38]) to the one-dimensional case. The number of particles will be given by a Bose-distribution integral with zero chemical potential,

$$N = \frac{L}{2\pi\hbar} \int \frac{dp}{\exp \varepsilon/T - 1} = \frac{L\sqrt{m^*T}}{\sqrt{8}\pi\hbar} \int_{\varepsilon_{min}/T}^{\infty} \frac{dz}{\sqrt{z}(\exp z - 1)}, \quad (1)$$

where $L$ is the length of the sample. Note that, in contrast to the 3D case, the integral diverges at low energies as $z^{-1/2}$, and the integration should start from the cutoff $\varepsilon_{min}/T$ where $\varepsilon_{min} = \hbar^2(2\pi/L)^2/(2m^*)$ with $2\pi\hbar/L$ corresponding to the minimum possible momentum. The integral in (1) can be then estimated as $\sqrt{T/\varepsilon_{min}} \simeq \sqrt{2m^*T}L/(2\pi\hbar)$, and we find

$$T^* \simeq \frac{4\pi^2\hbar^2 n_{vac}}{m^* L}, \quad (2)$$

where $n_{vac} \equiv N/L$ is the linear density of vacancies. For the 2/5 solid shown in Fig. 4b, $n_{vac} = 1/(3a) \approx 2.3$ nm$^{-1}$. With the length of our system $L = 700$ nm, the condensation temperature would be of the order of 10 mK depending on the effective mass $m^*$ of the vacancies.

The crystal with Bose-condensed vacancies would possess the so-called supersolid behavior, i.e., a superflow of mass, and at the same time the translational invariance, which would be an unknown phase of matter combining the properties of crystal and of superfluid at the same time[5–7].

To summarize, we have observed an unknown state of matter: a bosonic solid with mobile zero-point vacancies. This soft, mobile solid was identified as 2/5 commensurate phase consisting of bosonic dimers and containing significant amount of irremovable vacancies due to topological frustration. Due to the different inner pressures, two commensurate phases cannot coexist which we demonstrate experimentally by observing abrupt disappearence of the 1/3 phase once the 2/5 appears at corresponding filling. The mobile solid is thus realized in the 2/5 phase in contact with liquid or incommensurate solid which may tune their pressure by changing density, the freedom which commensurate phase does not have. The observed quantum state is truly extraordinary because, in contrast to Cooper pairs in superfluids and superconductors stabilized by phonons, pairing of

fermionic $^3He$ atoms on CNT is promoted by carbon lattice. The vacancies in such solid are delocalized below ∼0.1 K making a precursor of the supersolid which possesses at the same time crystalline symmetry and superfluid non-dissipative flow of mass. Extremely intriguing would be to study the nuclear spin system in the dimer solid phase because this phase, at an arbitrary mismatch, generally contains also significant amount of unpaired fermionic $^3He$ atoms. Pairing and superfluidity of the additional atoms is expected as well at the lowest temperatures, making the ensemble of $^3He$ atoms on a nanotube one of the richest available solid state systems, holding promise for supersolid and superfluid inside single nanoscale sample.

## Methods
### Sample preparation and calibration
The CNTs were synthesized in the gas phase with the floating catalyst chemical vapour deposition growth method (FC-CVD) followed by the direct thermophoretic deposition onto prefabricated chips[39]. A suspended nanotube can be viewed as a doubly clamped string with the fundamental oscillation frequency $F = (1/2\pi)\sqrt{k/m}$ where the spring constant $k$ is determined by the line tension of the tube, controlled by the DC gate voltage. The oscillation spectra were obtained with a frequency modulation method[40]. Temperature was measured using RuO$_2$ resistance thermometer from Lake Shore Cryotronics[41] and by a noise thermometer from Physical-Technical Institute (Physikalisch-Technische Bundesanstalt) in Berlin[42].

The mechanical oscillations of the tube are driven by an oscillating electrical force due to applied AC voltage between the tube and the gate. Helium atoms provide an additional mass $\delta m$ and an additional spring constant $\delta k$ for the nanotube (due to change in tension[27,43]). The relative frequency shift $\delta F/F = \delta k/2k - \delta m/2m$ of the mechanical resonance of the tube thus provides information of the involved energies and the nature of the phases formed by the adsorbed helium. The coverage $\rho$, defined as a particle density of helium atoms projected onto the carbon lattice, was calculated from additional mass $\delta m = -2m \delta F/F$ measured in the liquid phase at 0.25 K where we assume negligible additional stiffness $\delta k$. We have measured our spectra with different bias and RF voltages and did not observe any systematic dependence of resonant frequency neither on the Joule heating nor mechanical drive.

### Quantum diffusion at low densities
$^3He$ atoms on the nanotube lattice are attracted to the centers of carbon hexagons. The energy barrier separating two neighboring hexagon centers located at the distance of $d = \sqrt{3}a = 2.46$ Å (in the planar graphite/grafoil geometry) is quite large, $U = 2.7$ meV = 30 K[29]. The hopping time $t_h$ to the neighboring hexagon is $t_h \sim 1/(fP)$ where $P$ is the probability of tunneling through the barrier, and $f$ is the attempt frequency. At temperatures as low as 0.1 K, the probability of thermally activated tunneling $P_{th} \sim \exp -U/(k_B T)$ is negligible, and hopping is governed by the quantum tunneling which probability can be estimated in the quasi-classical approximation as $P_q \sim \exp -\sqrt{2m(U - E)}d/\hbar$ where $E$ is the lowest energy level in the potential well[38]. Neglecting the curvature of nanotube and taking the main Fourier term $U(x) = (U/2)(1 - \cos 2\pi x/d)$ for the shape of the potential, we obtain $U(d) \approx (\pi^2/d^2)Ux^2$ for its harmonic approximation. This yields for the oscillation frequency $f = \sqrt{U/(2md^2)} \sim 10^{12}$ 1/s and $E = \pi\hbar f \approx 20$ K for the lowest energy level. Consequently, the width of the barrier decreases drastically to $d' \approx 0.9$ Å, and the quantum tunneling probability becomes as large as $P_q \sim 0.4$.

We must note that the formula for $P_q$ in the previous paragraph is quasi-classical and accurate only if the exponent is much smaller than unity ($\hbar \to 0$); thus we can only conclude that $P_q$ is of the order of 1. The diffusion coefficient in the limit of small density can be crudely estimated as for the 2D random walk, $D = (1/2)d^2 f P_q \sim 10^{-8}$ m$^2$/s. Hence, we obtain a characteristic time scale $\tau_d \sim (1/4)L^2/D \sim 10$ μs of the diffusion of

a helium atom across the tube length of $L = 700$ nm. The characteristic time of redistribution of atoms over the tube is thus much longer than the period of oscillations of the tube, $1/F_0 \simeq 3$ ns, and therefore, configuration of helium remains unchanged during the measurements. The calculated diffusion time $\tau_d$ coincides with the time of ballistic flight of ground-state atoms with $k = \pi/L$[44] along the tube, $\tau_b = m_3 L^2/\pi\hbar = 8$ μs. This non-intuitive result agrees perfectly with the accurate theoretical calculations of the band structure model by Hagen et al. who concluded that the tunneling is so fast that there is only a little correction to a free particle motion[45].

The odd coincidence between predictions of the two completely opposite models, diffusion and free particle, is in fact not accidental. Indeed, the hopping time is connected with the length $d$ of one jump as $t_h = 1/\omega \sim md^2/\hbar$ if the probability $P$ of tunneling is of order 1. The diffusion time, onwards, scales with the length $L$ as $\tau_d \sim t_h(L^2/d^2) \sim mL^2/\hbar$, upto the limit $d \sim L$, where it becomes exactly the time of flight of a free particle with $k \sim 1/L$. This is a beautiful quantum size effect where diffusion time of particle depends only on the length of the sample ignoring any details of the corrugation potential. This scaling law is valid in higher dimensions as well. The view of the fast tunneling and diffusion of helium atoms described here is a property of an isolated atom and can be applied to low coverages only. At higher densities the attraction between atoms takes place, and dimerized solid becomes possible, as it is discussed in the Main text.

## 2D van der Waals approach

A pressure $P$ in two dimensions is naturally defined as the normal force per unit length of the boundary, arising from the kinetic energy of particles confined to in-plane motion on the CNT. The equation for ideal gas is identical to the 3D formula, $P = m(N/S)(\overline{v_x^2} + \overline{v_y^2}) = (N_a/S_m)k_B T = RT/S_m$, where the surface area $S_m$ of one mole plays the role of the molar volume in 3D, and $\overline{v_i^2}$ describes the average of square velocity of particles in the direction $i = \{x, y\}$. The van der Waals (vdW) correction term $b$ to the molar surface area $S_m$ is a constant equal to the total area occupied by the hard-cores of one mole of particles. Particles feel an attraction to the rest of the particles, which is proportional to the coverage $\rho \propto 1/S_m$. This leads to a change in the pressure at the boundary, but since the deficit of pressure is proportional to the number of particles, the correction is quadratic in coverage, $P \to P + a/S_m^2$. The functional form of the vdW equation is therefore the same as in 3D, but with the replacement of molar volume with molar surface area, $(P + a/S_m^2)(S_m - b) = RT$. The reduced form reads then universally, $\left[P/P_{cr} + 3(\rho/\rho_{cr})^2\right] \cdot (3\rho_{cr}/\rho - 1) = 8(T/T_{cr})$, where $T_{cr}$, $\rho_{cr}$, $P_{cr}$ are the critical values for temperature, density and pressure.

The above vdW form can also be applied to 2D $^3$He, a fermionic quantum system. The ideal gas law for fermions contains additional $\rho^2$ terms, while the exchange interaction yields an additional increase in pressure. However, both modifications can be included in the parameters of the vdW equation. A fit of the 2D quantum vdW equation to the transition temperature with $T_{cr} = 0.52$ K and $3\rho_{cr} = 4.2$ nm$^{-2}$ is shown in Fig. 2.

The validity of the vdW approach was also confirmed by calculating the frequency drop at the phase separation curve. We assume that liquid helium clusters start to form near the ends of the tube where the atomic binding energy is larger due to a surface roughness at the ends of nanotube[30]. Redistribution of $^3$He atoms from liquid puddles near the ends to a uniform fluid layer over the tube at the phase transition changes the resonant frequency according to

$$\frac{\delta F}{F_0} = \left[-\frac{1}{6\pi}\sin\pi\frac{\rho}{\rho_0} + \frac{1}{48\pi}\sin 2\pi\frac{\rho}{\rho_0}\right]\frac{\rho_0}{\rho_C}, \quad (3)$$

where $\rho_0 = 3\rho_{cr} = 4.2$ nm$^{-2}$, and $\rho_C = 38.2$ nm$^{-2}$ is the areal density of carbon atoms. The frequency jump calculated from Eq. (3), plotted in

Fig. 3 as a solid curve, fits the experimental data points very well. Note that there is no fitting parameter in Eq. (3), once the critical coverage $\rho_{cr} = 1.4$ nm$^{-2}$ is determined from the phase diagram (Fig. 2).

The largest liquid–gas coexistence coverage 4.2 nm$^{-2}$ is five times larger than the corresponding coverage in $^3$He on graphite, 0.8 nm$^{-2}$ [30]. This difference can be attributed to transverse, vertical motion of $^3$He on the nanotube. In the variational Monte-Carlo study on $^3$He monolayer on graphite there is no self-binding for a strict 2D system, but accounting for the vertical motion of atoms shows that $^3$He film forms a clustered liquid with critical coverage of $\approx 2$ nm$^{-2}$ [46].

## Solid helium on CNT

On graphite/grafoil, the first, 1/3 commensurate solid helium phase (Fig. 2) is formed at densities $4.3 < \rho < 7.1$ nm$^{-2}$ [13,14,16,18]. Dominating excitations in 1/3 solid on grafoil are phonons with heat capacity $C \propto T^2$ at low temrperatures[13,33]. In the case of helium adsorbed on nanotube, thermodynamics of 1/3 solid is guided by the 1D phonons, as orbital motion is frozen below 1 K (see main text). Free energy and other thermodynamical quantities can be calculated similarly to 3D case[47]:

$$
\begin{aligned}
F &= \frac{LT}{2\pi c}\int_0^\infty \ln(1 - e^{-\hbar\omega/T})\,d\omega = \frac{LT^2}{2\pi\hbar c}\int_0^\infty \ln(1 - e^{-z})\,dz \\
&= -\frac{LT^2}{2\pi\hbar c}\int_0^\infty \frac{z\,dz}{e^z - 1} = -\frac{\pi}{12}\frac{LT^2}{\hbar c}, \\
S &= -\frac{dF}{dT} = \frac{\pi}{6}\frac{LT}{\hbar c}, \quad P = -\frac{\partial F}{\partial L}\Big|_T = \frac{\pi}{12}\frac{T^2}{\hbar c}.
\end{aligned}
\quad (4)
$$

The second commensurate 2/5 dimer solid (Fig. 3) has been observed on graphite in the narrow coverage range $7.1 < \rho < 7.8$ nm$^{-2}$ [18,33]. At higher densities, the dimer phase is energetically unfavorable in the planar geometry, because the hard-core diameter of helium $\sigma_{hc} = 2.65$ Å[21] is larger than the distance $L_{gr} = 2.46$ Å between neighboring cites, and the incommensurate solid structure appears instead[14,16,33]. However, on a curved nanotube the distance between neighboring cites in the direction perpendicular to the tube's axis is larger than on graphite/grafoil, which admits dimers at large densities of a sub-monolayer. For our tube of radius $r = 8$ Å, the distance between atoms in a dimer perpendicular to the tube's axis is about 30% larger than in the plane geometry[31], $L_{CNT,\perp} \simeq 3.2$ Å $> \sigma_{hc}$. The 2/5 solid phase consisting of such dimers [Fig. 4b] is therefore admitted also at high coverages. Furthermore, the distance between atoms in a perpendicular dimer on CNT is very close to He-He potential minimum at 3.0 Å[21]. Incommensurate phases on nanotube are energetically unfavorable as compared to the commensurate 2/5 solid, because the latter makes use of all available potential minima provided by the carbon lattice and He-He interaction.

The diffusion Monte Carlo simulations by Gordillo and Boronat, however, show that the energy of the incommensurate phase on the nanotube is slightly, within a fraction of percent, lower than that of the 2/5 phase[31]. However, they considered only nanotubes of the so-called armchair type ([NN] in standard notations) which does not allow dimers perpendicular to the axis. The dimers oriented parallel to the axis indeed have large energies due to the short-range repulsion, as was explained above. Our tube was definitely not of the armchair type but rather chiral semi-metallic with the room temperature resistance of 100 kOhm, while the pure armchair tubes are metallic with the order of magnitude smaller resistance. Incommensurate solid is not compatible with the properties of helium on CNT at coverages above 6.7 nm$^{-2}$. The observed excitation spectra are linear [see Fig. 4a], in line with the model of weakly interacting vacancies, rather than with phonons in incommensurate solid. Furthermore, at coverages >8 nm$^{-2}$ and $T > 0.1$ K the abrupt appearance of the additional tension is observed (Figs. 1f and 2). This stiffening, which we attribute to localization of vacancies, cannot be explained by pinning of helium lattice which might occur with lowering temperature and would have a shape of a smooth crossover

rather than the observed sharp transition. Furthermore, pinning-depinning transition temperature should be drastically sensitive to amplitude of the oscillations, while we did not observe any dependence on the RF drive in the power range −40...−65 dBm.

Another possible commensurate phase might consist of rows perpendicular to the axis of CNT, like the 1/2 phase suggested by Lueking and Cole[48]. However, the rows must be unstable due to the large zero-point motion. Indeed, each helium atom-in-a-row is localized in the potential well with almost vertical walls due to the short-range repulsion from neighboring atoms. The width of the potential well for the atom between them is $\lambda/2 \simeq 2(L_{CNT,\perp} - \sigma_{hc}) = 1.1\,\text{Å}$ which gives zero-point energy $E_{zp} = 2\pi^2\hbar^2/(m_3\lambda^2) \simeq 70\,\text{K}$. In a dimer, zero-point motion is the ground state of the vibrational and rotational degrees of freedom with energies $E_{v,0} \sim E_{r,0} \sim \pi\hbar f \sim 20\,\text{K}$ per atom. The ground state energy of atom in the dimer is thus $E_0^{dimer} = E_v + E_r \simeq 40\,\text{K}$ above the adsorption energy, a value much smaller than that of the atom-in-a-row configuration.

There is a basic question: how phases are arranged on the nanotube in the general case when the coverage is arbitrary but not exactly that of the 1/3 or the 2/5 commensurate phases. First, we note that two different commensurate solids cannot co-exist on CNT nor on graphite/grafoil, although many authors have suggested a domain wall structure which incorporates different commensurate phases[16,33,49]. The prohibition of existence of more than one commensurate phase on a surface of an adsorbent is provided by the fact that, at a given temperature, pressures of different commensurate phases are unique and different intrinsically. The uniqueness of the pressure follows directly from the circumstance that density of commensurate solid is set by the underlying carbon lattice, and thus cannot vary. In the 3D case, the situation is completely different: two crystals with different structures may coexist because the pressures in two bulk solids can be varied by changing densities. In contrast, in the case of the adsorbed particles, the density of commensurate phase cannot be varied but fixed by the carbon lattice.

On the grounds of the reasoning given above, we can propose which phases are realized at particular densities of helium on a CNT. Below the coverage of the 1/3 commensurate phase, $\rho_{1/3} = 6.4\,\text{nm}^{-2}$, a coexistence of the 1/3 phase and liquid[14,16] takes place. Once the filling exceeds that of the 1/3 phase, it is replaced by a mixture of the 2/5 phase and liquid. Since pressure of the 2/5 solid differs from that of the 1/3 solid, equilibrium between these phases is not possible, and hence the 1/3→2/5 quantum phase transition is extremely sharp, as one can see in Figs. 1c, d, 3 and Fig. 4a. The transition occurs at the coverage 6.7 nm$^{-2}$ just above that of the 1/3 solid (Fig. 3). At densities larger than that of the 2/5 phase, $\rho_{2/5} = 7.6\,\text{nm}^{-2}$, coexistence of the 2/5 phase and incommensurate solid phase is realized.

The existence of the commensurate solid at all densities exceeding $\rho_{2/5} = 7.6\,\text{nm}^{-2}$ upto full first layer filling is thereby confirmed by the pressure measurements as a function of temperature, Fig. 4a. Incommensurate solid on graphite has a 2D phonon excitation spectrum with quadratic heat capacity, $C \propto T^2$ [33]. In the 1D case of a nanotube, where short-length transverse phonons are frozen, the heat capacity of incommensurate phase must be linear, and the pressure must be quadratic with temperature. Instead, we have measured linear dependence $P(T)$ in the low-temperature phase on CNT at high densities, Fig. 1a, resembling excitation spectrum of the ideal gas. As the pressure in the commensurate phase is a function of temperature only, pressure in a coexisting incommensurate phase is tuned to be the same by changing its density with changing temperature. A free parameter allowing the change in density of incommensurate phase is relative length occupied by the incommensurate solid on the tube.

### Ground state of a dimer
Similarly to diatomic molecules, dimers in the 2/5 commensurate solid have vibrational and rotational mechanical degrees of freedom. In a typical diatomic molecule in the electronic ground state $A^1\Sigma_g^+$ the

oscillation energy is by two-three orders of magnitude larger than the rotational one[38], but our dimers are special as they are stabilized foremost by the interaction with the underlying carbon atoms. The vibrational energy can be estimated using harmonic approximation of the potential induced by carbon lattice, $E_v = 4\pi\hbar(v + 1/2)f$ where $f \sim 10^{12}\,\text{Hz}$ is the oscillation frequency estimated above. The quantum $\Delta E_v \sim 80\,\text{K}$ is much larger than the solidification temperature, and the vibrational motion of dimers is frozen.

The free rotation of dimers is prohibited in the solid phase because of the repulsive interaction with the nearest dimers. Nevertheless, there is a peculiar rotational-vibrational mechanical mode in which helium atoms oscillate in antiphase in the direction perpendicular to the dimer axis. The orbital momentum of electrons is zero, and thus the rotational-vibrational ground state corresponds to $J = 0$ (see for instance[38]), while the first excited level with $J = 1$ has an energy similar to vibrational quantum $\Delta E_v \sim 80\,\text{K}$ and, therefore, all dimers at temperatures around 0.1 K are in the symmetric ground state. Moreover, the freezing of the rotational degree of freedom in diatomic molecule stabilizes the nuclear spins to the state $S = 0$. This is a beautiful, combined quantum mechanical and symmetry effect well-known from ortho-$H_2$ and para-$H_2$ problem[50].

The total wavefunction of the diatomic molecule having two identical fermionic nuclei must by antisymmetric with respect to particle permutation. In the para-molecule nuclear spins are anti-parallel, $S = 0$, and the wavefunction of nuclei is antisymmetric with respect to particle permutation, and the rotational wavefunction must be thus symmetric with even $J = 0$, 2, 4... In the ortho-molecule nuclear spins are parallel, $S = 1$, and the nuclear wavefunction is symmetric, then the rotational wavefunction must be antisymmetric with odd quantum number $J = 1$, 3, 5...[50]. In contrast to the almost spherical molecule of hydrogen, helium dimer cannot rotate freely in the solid phases. Nevertheless, the angular vibration described above obeys a similar selection rule. The ground rotational-vibrational state is symmetric with respect to the inversion and, therefore, the nuclear spin wavefunction must be anti-symmetric with anti-parallel spins, $S = 0$. Thus, despite the extreme weakness of both dipole-dipole interaction of $^3$He nuclei, $\varepsilon_{d\text{-}d} \sim 0.1\,\mu\text{K}$, and exchange interaction, $I \sim 1\,\text{mK}$[51], the ground state with $S = 0$, $J = 0$ of $^3$He dimer is protected, due to symmetry, by a much larger rotational vibration quantum. All dimers are thus in the identical (ground) states, which allows for the resonant tunneling of topologically stabilized vacancies.

## Data availability
The data that support the findings of this study are available from the corresponding authors upon request.

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

## Acknowledgements

We are grateful to Henri Godfrin, Ari Harju, Ville Havu, Andreas Huettel, Martti Puska, and Erkki Thuneberg for discussions, and Petri Tonteri from Densiq Ltd. (90620 Oulu, Finland) for providing us with the ultra-pure grafoil. We thank Miika Haataja for measuring the surface area of the grafoil sample and Qiang Zhang for participation in the optimization of the CNT process. **Funding:** This work was supported by Academy of Finland projects No. 314448 (BOLOSE), No. 312295 (CoE, Quantum Technology Finland), and No. 316572 (CNTstress). The research leading to these results has received funding from the European Union's Horizon 2020 Research and Innovation Programme, under Grant Agreement No 824109 (EMP), under ERC Grant No. 670743 (QuDeT), and in part by Marie-Curie training network project (OMT, No. 722923). J.-P.K. is grateful for the financial support from Vilho, Yrjö and Kalle Väisälä Foundation of the Finnish Academy of Science and Letters. This research project utilized the Aalto University OtaNano/LTL infrastructure.

## Author contributions

I.T., M.K., and A.S. conducted the experiments with the help of E.S. in cryogenic matters. The data analysis was performed by I.T. together with

contributions from M.K., E.S., and P.JH. The sample fabrication and its process development was done by J-P.K., T.S.A., and Y.L. Initial characterizations were carried out by M.W. The manuscript was written by I.T., M.K., A.S., and P.J.H. with the help of comments from all authors. The work was supervised by E.K. and P.J.H.

## Competing interests

The authors declare no competing interests.
