## [Peer Review File · Nature Communications]

Topologically-imposed vacancies and mobile solid ^3He on carbon nanotubeREVIEWER COMMENTS

Reviewer #1 (Remarks to the Author):

This paper presents significant and interesting new results on the unusual behavior of submonolayer ^3He films adsorbed on a carbon nanotube (CNT). The measurements clearly show transitions between different ^3He phases, at temperatures as low as 20 mK. The results suggest that one of the phases is a fluid-like “gliding solid” and the authors interpret this as evidence for a dimer phase with intrinsic zero point vacancies (due to the finite circumference of the graphene sheet when rolled to make the nanotube). The discovery of such a new phase in helium films, and the possibility of creating a 2D supersolid phase on nanotubes, would definitely justify publication in Nature Communications.

The behavior of helium films has been studied for many years and includes the discovery of fundamental phase transitions like the Kosterlitz-Thouless vortex unbinding transition. The availability of high quality graphite substrates allowed well-characterized 2D layered systems to be studied. These included fluid (liquid and gas) layers and registered (commensurate) solid phases, of which the $1/3$ phase is well established. The two isotopes of helium (^4He and ^3He) allow quantum effects to be controlled and studied (bose/fermi, spin, different masses/zero point motion...). However, substrates like grafoil are not perfect, and the interaction between helium atoms and the graphite substrate is quite strong, so many of the interesting phases only exist in multilayer films (e.g. in the 2nd or 3rd layer of helium) where interactions between helium atoms in different layers can complicate the study of specific phases. The current state of experiments on helium films on graphite was summarized in a recent review (Ref. 17) that points out how graphene and carbon nanotubes provide promising new substrates on which to study the behavior of helium films. Such substrates can have very high quality and their interactions with helium atoms are much weaker than on bulk graphite. This favors highly quantum behavior in the first layer, in contrast to other systems in which the helium is tightly bound to the substrate. CNTs are particularly attractive as substrates for sensitive measurements on helium films since a CNT can be used as the vibrational element of a mechanical resonator, and is sensitive to both submonolayer helium masses and surface tension changes due to the adsorbed helium atoms.

The current paper presents the first detailed study of submonolayer ^3He films adsorbed on a CNT resonator. It clearly demonstrates the method's potential and reveals significant differences from previously studied ^3He films on graphite. The only other comparable experiment I am aware of involved multilayer ^4He films on a CNT resonator by Noury et al. (Ref. 27). That work confirmed the sensitivities to both mass and to surface tension that are exploited in the current work. It also confirmed the high quality of the CNT surfaces, which clearly showed a whole series of layering transitions. However, the behavior of ^4He films is not nearly as rich and complicated as that of ^3He films.

The present work reveals some of that richness. The fact that the ^3He films are submonolayer makes the interpretation of phases clearer than in multilayer films on other substrates. The authors were able to clearly identify the fluid (gas/liquid) phase transitions (via the mass sensitivity) and to calibrate their coverage scale. They could then confirm the existence of the expected commensurate $1/3$ solid phase (one helium atom for every three carbon sites). This phase is well established for ^3He (or ^4He) on grafoil, but with higher melting temperatures because of the larger binding potentials. Together, the authors' identification of these phases on their CNTs gives a clear starting point to study transitions to (and the properties of) other solid phases in the ^3He film. In the present experiments on a CNT, there is a transition to a higher density phase, and the situation is clearer than it would be in multilayer films on grafoil.

The central result of this paper (in addition to the proposed dimer structure) is the experimental evidence for a “freely gliding” $2/5$ solid phase that moves freely with respect to the CNT surface. The evidence for this is the fact that the CNT resonance frequency with this phase is essentially the same as for a liquid phase, and is significantly lower than in the lower density $1/3$ phase. Since the behavior

extends to very low temperature, it is identified as a quantum phase transition.

The connection between CNT oscillation frequency and mobility of the solid phases isn't initially obvious, since it is not due to mass decoupling but rather to changes in the CNT's line/surface tension depending on where/whether the helium atoms are localized with respect to the CNT carbon structure. However, the general effect is clear (and is well known to experts on nanotubes or graphene, e.g. in the ^4He work on CNTs).

The authors' interpretation of gliding in the proposed $2/5$ phase is carefully discussed in their "methods" section and is plausible, although other helium structures had to be considered. They emphasize the important point that rolling up a perfect graphene sheet to make a tube introduces a finite concentration of non-thermal defects into the ^3He . A mismatch effectively introduces extra carbon sites that would behave as vacancies in the helium film. These vacancies have essentially fixed concentration and so would be present even at zero temperature. They would behave as an ideal gas and would be a candidate for the supersolid state that was unsuccessfully sought in bulk ^4He (where there do not appear to be any zero point vacancies). Supersolidity requires that the particles (vacancies) be bosons, which would only be possible if the ^3He fermion atoms were organized as dimers, analogous to the paired ^3He atoms responsible for superfluidity in liquid ^3He at mK temperatures.

The CNT measurements are also able to probe a thermodynamic property of the helium films since the pressure in the films acts as a tension and changes the CNT oscillation frequency. They exploit this by measuring the temperature dependence of the frequency/pressure at low temperatures. Since the temperature dependence depends on both dimensionality and the type of thermal excitations, they have another probe of the state of the helium films. A change from quadratic to linear behavior in going from the $1/3$ solid to the gliding solid supports their interpretation of vacancies in the dimer phase.

The experiments described in this paper are innovative and challenging. The results are intriguing and they confirm the expectation that helium films on CNTs will open up an exciting new area of research. The mobility of the new solid phase is strong evidence of new physics in solid ^3He films. The interpretation in terms of a ^3He dimer phase with zero point vacancies is carefully explained and is plausible although the lack of direct structural information on the new phase leaves some uncertainty. This work represents a significant step in creating new quantum states in helium films and measuring their mechanical properties, another example of helium's long tradition of providing unique quantum states and fundamental insights.

The paper is well written and its length and style are suitable for this journal. I found almost no typos or similar errors to correct.

I strongly recommend that the paper be published in Nature Communications, in its present form.

Reviewer #2 (Remarks to the Author):

The manuscript presents a very interesting experimental study of the mechanical properties of carbon nanotubes covered by He^3 atoms. In a specific range of He^3 concentration, the system shows no mechanical resonance frequency shift with respect to the liquid phase. The suggested explanation of this effect is the presence of mobile dislocations in the so-called $5/2$ dimer phase of helium atoms absorbed on the carbon surface. This conclusion is supported by the temperature dependence of the mechanical resonance frequency which appears to be linear indicating that the system hosts the temperature-independence number of vacancies.

The results are interesting and deserve to be published in Nature Communications. There is however an important question about the role of topology which needs to be clarified. From the text, it seems that the term "topologically protected" is used to highlight the constant number of dimer vacancies in

5/2 phase determined just by the geometry of the system. But in fact, this is not enough. There should be some topological invariant which classifies the defects and explains their stability. This aspect is clearly missing in the manuscript. Before recommending the paper for publication I would like to ask the authors to improve or adjust the presentation accordingly.

Reviewer #3 (Remarks to the Author):

The paper presents results of the experimental investigation of suspended carbon nanotubes (CNT) with absorbed He atoms. The phase diagram of absorbed helium atoms is determined from measurements of frequencies of mechanical oscillations of CNT. The authors revealed a new state of the absorbed helium, which was identified as a 2/5 commensurate solid consisting of bosonic dimers of He atoms and containing significant amount of mobile bosonic dislocations.

The new state is interesting as a material for the search of the phenomenon of supersolidity possible in the BEC of dislocations. The author estimated the BEC temperature for dislocations in the new state. Although this temperature was not reached in their experiment it is not too much less than the temperatures already available.

I think that the paper is quite interesting, and I support its publication.

We are grateful to the Referees for their positive evaluation of our work, and for very useful comments and suggestions.

We accept all suggestions and, accordingly, make the following changes to the original submission.

Reviewer #2:

The results are interesting and deserve to be published in Nature Communications. There is however an important question about the role of topology which needs to be clarified. From the text, it seems that the term "topologically protected" is used to highlight the constant number of dimer vacancies in 5/2 phase determined just by the geometry of the system. But in fact, this is not enough. There should be some topological invariant which classifies the defects and explains their stability.

Our response:

1. We agree that the term "topologically protected solid" is not fully exact and may be misleading. Therefore, we have changed the title to "Topologically-imposed vacancies and mobile solid ^3He on carbon nanotube" which is more correct than the original "Topologically protected mobile solid ^3He on carbon nanotube".

2. Concerning the question about topological invariant, we have added a footnote

"From the topological point of view, the vacancies are stabilized by the discrepancy between chirality $[N M]$ of the nanotube, and the translational vector $[n m]$ of the particular helium lattice. For the simplest case of the $1/3$ commensurate solid with translations $t_1=[1 1]$ and $t_2=[1 -1]$, the discrepancy $D = \min(|N M - i*t_1 - j*t_2|)$ can be considered as the topological invariant. Indeed, the chirality of the tube cannot be changed by the deformation only, without cutting/glueing. Hence, the invariant D which determines and conserves the number of zero-point defects cannot be changed due to the mutual motion of carbon and helium lattices. The same is valid for any commensurate helium solid; for instance, $2/5$ has $t_1 = [2 1]$ and $t_2 = [2 -1]$, and has the discrepancy $D = [0 2]$ with the semi-metallic, nearly armchair tube $[22 1]$. Such a discrepancy provides and stabilizes two single vacancies per one carbon ring, or one dimer vacancy."

to the top paragraph on Page 14.

REVIEWERS' COMMENTS

Reviewer #2 (Remarks to the Author):

In their reply to my comments, the authors have explained the role of topology in the formation of dislocations. They also provided a footnote in the manuscript clarifying the structure of the topological invariant that describes the number of zero-point defects. I find these explanations comprehensive and can now recommend the paper for publication.